# Effects of implementation of a care bundle on rates of necrotising enterocolitis and own mother's milk feeding in the East Midlands: protocol for a mixed methods impact and process evaluation study

Janine Abramson,[1,2] Lisa Szatkowski [1,2] Manpreet Bains [2] Elizabeth Orton,[2] Helen Budge,[1,1,2] Marie Spruce,[3] Shalini Ojha [1,2,4]

¹Centre for Perinatal Research, School of Medicine, University of Nottingham, Nottingham, UK
²Lifespan and Population Health, School of Medicine, University of Nottingham, Nottingham, UK
³NEC UK, Nottingham, UK
⁴Neonatal Unit, University Hospitals of Derby and Burton NHS Foundation Trust, Derby, UK

**Correspondence to**
Professor Shalini Ojha;
shalini.ojha@nottingham.ac.uk

## ABSTRACT

**Introduction** Prevention of necrotising enterocolitis (NEC) is vital for improving neonatal outcomes. Feeding own mother's milk helps prevent NEC. Rates of own mother's milk feeding in the East Midlands are lower than the national average and the incidence of NEC is higher. The East Midlands Neonatal Operational Delivery Network (EMNODN) has created a care bundle to improve these in babies born at <32 weeks' gestation, the group at the highest risk of NEC. The bundle was introduced in September 2022 and embedded by December 2022. We will evaluate its effectiveness and conduct a process evaluation to understand barriers and facilitators to implementation.

**Methods and analysis** We will conduct a retrospective cohort study (workstream 1) using data from the National Neonatal Research Database (NNRD). We will identify infants receiving any own mother's milk on day 14 and at discharge, and cases of severe NEC. We will aggregate outcomes by birth month and use interrupted time series analysis to estimate an incidence rate ratio for changes after the care bundle was embedded, relative to pre-implementation. We will model data from all other NNRD units and assess whether there are any concurrent changes to exclude confounding due to other events. We will apply the RE-AIM framework (workstream 2), supplemented by the Consolidated Framework for Implementation Research and Framework for Implementation Fidelity, to conduct a mixed methods evaluation in EMNODN units. We will triangulate data from several sources, including questionnaires and semistructured interviews with parents and healthcare professionals, and data from patient records.

**Ethics and dissemination** The study has approval from the South East Scotland Research Ethics Committee 01 and the Health Research Authority and Health and Care Research Wales (IRAS 323099). Results will be disseminated via scientific journals and conferences, to neonatal service commissioners and through public-facing infographics.

**Trial registration number** NCT05934123.

## STRENGTHS AND LIMITATIONS OF THIS STUDY

⇒ Use of data from the National Neonatal Research Database allows comparison of all infants admitted to neonatal units in the East Midlands with a control group of all infants admitted to units elsewhere in England and Wales.
⇒ Interrupted time series analysis will allow any changes in own mother's milk feeding and the incidence of necrotising enterocolitis (NEC) related to implementation of the care bundle to be identified over and above any underlying changing trends.
⇒ The parallel mixed methods process evaluation will allow us to unpack the 'how' and 'why' underpinning the results of the interrupted time series analysis.
⇒ Analysis of routine data will allow all eligible infants to participate without risk of under-recruitment from underserved groups. In the process evaluation, we will purposively seek to include parents from underserved groups who traditionally hesitate to participate in research.
⇒ NEC is a rare occurrence and so statistical power to detect small changes in incidence related to implementation of the care bundle may be limited.

## INTRODUCTION

Necrotising enterocolitis (NEC) is a serious, life-threatening disease in preterm infants.[1] An estimated 12% of infants born weighing <1500 g develop NEC and 30% of these die.[2] Survivors have increased risk of adverse neurodevelopmental outcomes, including cerebral palsy and cognitive impairment.[3] Encouraging and supporting own mother's milk feeding prevents NEC and has several additional advantages, including improved long-term neurodevelopment and parent–infant bonding.[4] Health economic analysis estimates that if all premature infants were to

be fed own mother's milk, the total lifetime cost saving to the National Health Service (NHS) would be £46.7 million (£30.1 million in the first year of life), and there would be 238 fewer deaths per year with a gain of £153.4 million in lifetime productivity.[5]

Neonatal critical care services in England are commissioned by NHS England and managed within 11 neonatal operational delivery networks, to ensure that babies and their families receive high-quality, equitable and accessible care. The East Midlands Neonatal Operational Delivery Network (EMNODN[6]) comprises 11 neonatal units across seven NHS trusts, covering an area of 15 627 km², a total population of approximately 4.9 million[7] and approximately 47 000 live births per year.[8] In 2022, EMNODN units provided 50 443 days of care in total to infants born preterm, or unwell at term, who needed specialist neonatal care; 476 infants were cared for who were born before 32 weeks' gestational age (GA) and survived at least 48 hours.[9]

The rate of NEC in the EMNODN has remained above the national average since 2017, when this measure was added to the National Neonatal Audit Programme (NNAP).[10] Additionally, there are wide variations between the 11 units that comprise the EMNODN. Rates of any breastmilk feeding at discharge from EMNODN units have been consistently below the national average, again with wide local variations.

The EMNODN has created a NEC care bundle[11] which was introduced to the 11 neonatal units in September 2022, with use fully embedded by December 2022. Care bundles are a group of evidence-based, simple-to-implement interventions related to a disease or care process that, when executed together, result in better outcomes than when implemented individually.[12] The EMNODN NEC care bundle consists of recommendations and parent-facing information to promote own mother's milk feeding and optimise nutritional care of preterm infants. Early start and continuation of feeding with own mother's milk such that babies go home breast feeding is vital to ensure babies continue to receive own mother's milk throughout neonatal care and remain at a reduced risk of NEC.

There is some evidence that quality improvement (QI) programmes using care bundles can improve rates of own mother's milk feeding in neonatal units. In the East of England, introduction of a care bundle increased the prevalence of own mother's milk feeding from 50% to 57%.[13] However, no previous studies have explored the effect of complex, multicomponent interventions such as a care bundle on rates of NEC and other feeding practices, and there are no published process evaluations of intervention implementation.

### Objectives

Our study comprises two parallel, integrated workstreams with the following objectives:

Workstream 1: Estimate the effects of implementation of the NEC care bundle on rates of own mother's milk feeding and incidence of NEC, comparing infants admitted to the 11 EMNODN neonatal units with admissions to all other units in England and Wales. We will also describe feeding practices before and after implementation of the care bundle and examine the infant, maternal and unit-level characteristics associated with own mother's milk feeding and NEC.

Workstream 2: In a multicentre, mixed methods process evaluation in the 11 EMNODN neonatal units, evaluate how well the care bundle was implemented, understand barriers and facilitators to implementation and understand which components contributed most to effectiveness.

## METHODS AND ANALYSIS
### Design

Parallel, integrated workstreams:

Workstream 1: Retrospective cohort study using routinely collected clinical data, with interrupted time series (ITS) analysis.

Workstream 2: Multicentre, mixed methods process evaluation based on the RE-AIM framework,[14 15] Consolidated Framework for Implementation Research (CFIR)[16] and Framework for Implementation Fidelity (FIF).[17]

### Setting

Workstream 1: Neonatal units in England and Wales, comparing infants admitted to the 11 neonatal units in the EMNODN with infants admitted to all other units.

Workstream 2: The 11 neonatal units that comprise the EMNODN.

### Time period

1 January 2016 to 30 September 2023, with infants grouped by date of birth into three groups according to date of implementation of the care bundle: pre-implementation period—1 January 2016 to 31 August 2022; implementation period—1 September 2022 to 31 December 2022; evaluation period—1 January 2023 to 30 September 2023, followed up for 3 months (until 31 December 2023) to cover the full neonatal stay of the final participant.

### Data source

Workstream 1: Routinely recorded electronic patient record data from all admissions to NHS neonatal units in England and Wales, held within the National Neonatal Research Database (NNRD). Data are entered by clinical staff at the point of care and extracted, anonymised and combined by the Neonatal Data Analysis Unit at Imperial College, London. The NNRD has NHS Research Ethics Committee approval (10/80803/151) and approval from the Caldicott Guardians of NHS trusts for research use.

Workstream 2: Quantitative and qualitative data from a variety of sources: questionnaires with parents and healthcare professionals (HCPs); semistructured one-to-one interviews with parents and HCPs; NNRD data, supplemented with data directly from paper-based and

electronic patient notes, collected using an electronic case report form, for data items not available within the NNRD; data from EMNODN units on cot capacity, staffing levels and facilities.

Questionnaires will be completed electronically using an online survey platform, with paper copies available should participants request them. Any data from questionnaires completed on paper will be manually digitised, with a sample (10%) checked for verification of data entry. Audio recordings of interviews will be transcribed verbatim and transcripts will be checked against audio recordings to ensure they are accurate and that any identifiable information discussed during interviews is anonymised.

## Eligibility criteria

For workstream 1, we will include infants who: were born at <32 weeks' GA; survived at least 48 hours from birth; and were admitted for neonatal care from 1 January 2016 to 30 September 2023 to a neonatal unit contributing data to the NNRD. In line with previous work,[18 19] we will exclude infants with missing data on key characteristics (eg, sex, GA, final discharge destination) as well as infants admitted >12 hours after birth and those missing data for one or more episodes of care.

For workstream 2, we will include parents of infants who: are admitted to one of the 11 EMNODN neonatal units in a 6-month period after implementation of the care bundle; are born at <32 weeks' GA; and survived at least 48 hours from birth. All eligible parents will be invited to respond to the questionnaires. For the interviews, parents will be purposively sampled to include: parents from different neonatal units; parents of extremely preterm infants (born at 22–27 weeks' GA) and very preterm infants (born at 28–32 weeks' GA); parents whose infants received own mother's milk, mixed feeding or did not receive any own mothers' milk; parents of infants who had suspected or confirmed NEC; and parents representing different maternal (eg, education) and environmental (including socioeconomic) factors.

We will recruit HCPs working in EMNODN neonatal units. All HCPs will be invited to respond to the questionnaires and some will be purposely sampled to include a range of individuals representing different job roles (unit leads, consultants, junior doctors, nurse practitioners, nurses, healthcare assistants and allied health professionals including dietitians, speech and language therapists) and hospital sites.

All questionnaire respondents and interview participants will provide written informed consent for their participation and use of the data provided, including deidentified direct quotations in study outputs.

## Intervention

The NEC care bundle, designed by a multidisciplinary team, consists of recommendations for neonatal units, alongside parent-facing information, to promote own mother's milk feeding and ensure optimum nutritional care of preterm infants. A schematic representation of the care bundle is shown in figure 1, giving an overview of an infant's journey through neonatal care and opportunities to reduce the risk of NEC.

Recommendations to optimise nutritional care include guidance on: ensuring early and adequate parenteral nutrition given when infants cannot have sufficient milk feeds such as in the first hours after birth, while building milk feeds and when unwell (following established National Institute for Health and Care Excellence guidelines[20]); use of oropharyngeal colostrum[21] (the practice of giving own mothers' first milk directly into the infant's oropharynx to enhance immunity and protect against NEC); optimal time and speed of increasing milk feeds[22]; and use of other interventions such as human donor milk,[23] probiotics[24] and breastmilk fortifiers.[25]

### Primary outcomes

► Receiving any own mother's milk at discharge (exclusively or in combination with infant formula).
► Receiving any own mother's milk on day 14 of life (exclusively or in combination with infant formula).
► Severe NEC requiring surgery or resulting in death.[26]

### Secondary outcomes

► Exclusive feeding with own mother's milk at discharge.
► Any NEC, including NEC not requiring surgery and not resulting in death.

### Data analysis strategy: workstream 1

After data cleaning, the number of infants with missing data across key variables will be tabulated and the total number of exclusions and final study population size determined. Descriptive statistics will be presented to summarise the distribution of baseline variables during each of the three study phases, separately for the EMNODN units (intervention group) and all other units in England and Wales (control group). Continuous variables will be reported with means and SDs, if shown to be normally distributed using a combined skewness and kurtosis test, or otherwise will be reported using medians and IQRs. Categorical variables will be reported with frequencies and percentages. P values for differences between the three time periods will be reported using the $\chi^2$ test or the Wilcoxon rank-sum test, as appropriate.

We will use ITS analysis to assess the impact of implementation of the care bundle on the outcomes listed above, as has been used previously in neonatal research.[27] This is a strong quasiexperimental design which accounts for underlying trends in an outcome. If, for example, rates of NEC were already in decline prior to the implementation of the bundle, the ITS methodology will detect any additional impact of the bundle, above the background decline. It can also account for the impact of any other interventions or changes in population characteristics which might themselves impact outcomes.

We will aggregate outcomes by month of birth (or by 2-month period if necessary to improve statistical power,

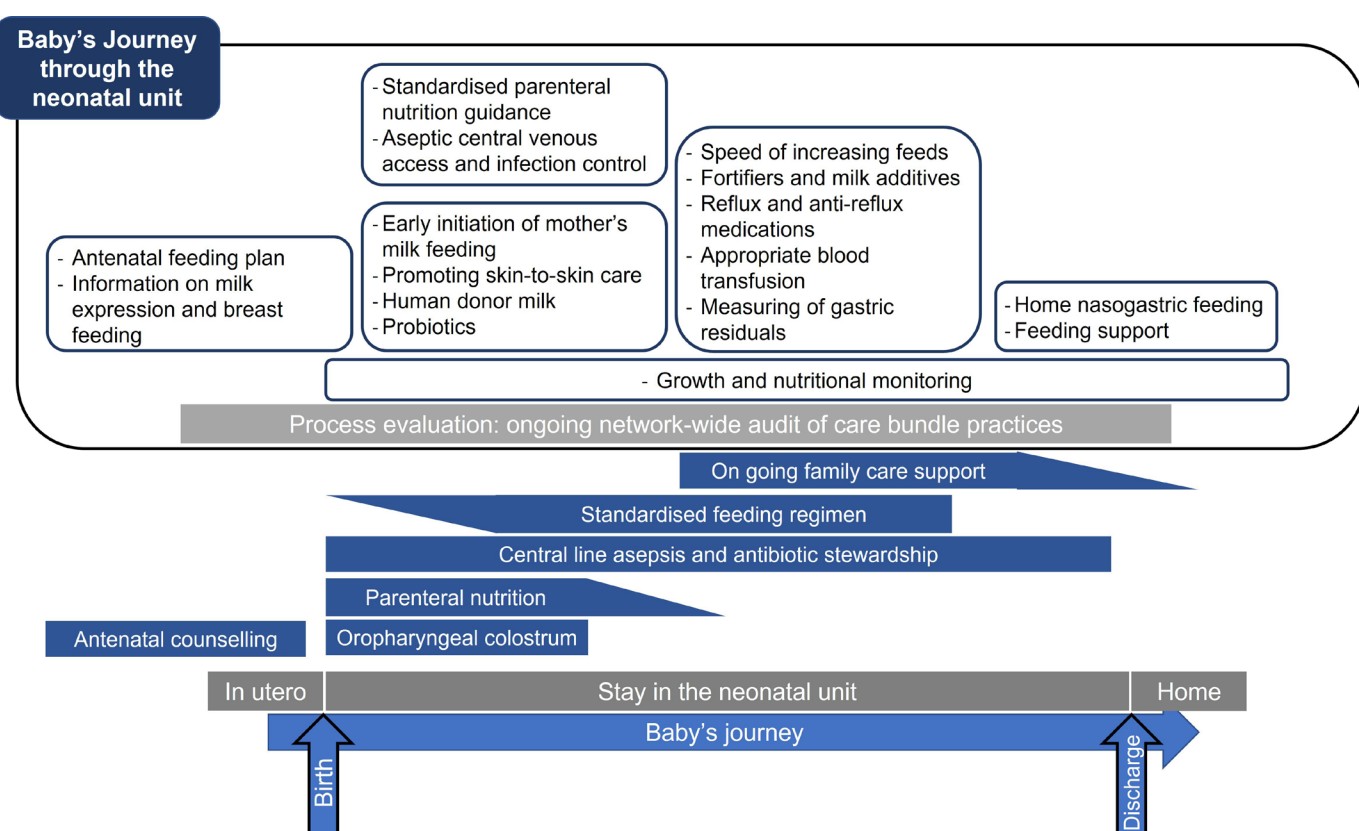

**Figure 1** Overview of the East Midlands Neonatal Operational Delivery Network (EMNODN) necrotising enterocolitis (NEC) care bundle.

without detriment to interpretation of results) and plot time series graphs to show the number of cases of each outcome per month over the study period in the EMNODN units combined. We will use a quasi-Poisson generalised additive mixed model[28] to estimate an incidence rate ratio (IRR) for changes in the incidence of the outcomes in EMNODN units during the evaluation phase, relative to the pre-implementation period. Data for the 4-month implementation phase will be plotted graphically but excluded from statistical modelling as this period is too short to be able to model changes in incidence compared with the pre-implementation period. We will similarly plot and model data for infants admitted to units in the control group, to assess whether there are any changes in the IRR to exclude confounding due to other events or interventions which occurred around the time that the care bundle was introduced.

We will incorporate several covariates into our models as follows:

► The total number of admissions per month will be used as an offset term in each model to account for variations in the number of babies 'at risk' of the outcomes.

► The secular trend in each outcome will be captured using a continuous variable coded from 1 to 93 (the total number of months of data across the study period).

► To assess whether there was a step change in the outcomes, we will include a binary variable coded as 0 during the pre-implementation phase and 1 during the evaluation phase.

► To assess whether there was any change in trend (upwards or downwards) in the outcomes, we will include a binary variable coded as 0 during the pre-implementation phase and from 1 to 9 (the number of months in the evaluation phase) during the evaluation phase. We will also assess whether there are any quadratic or cubic trends in the outcomes in the evaluation period, though the relatively short (9 months) length of this evaluation period will limit the ability to detect any non-linear trends. We will use the Akaike information criterion (AIC) to compare models with linear, quadratic and cubic trends in the evaluation period, with a smaller AIC indicating better model fit.

► Month of the year will be included as a cyclic cubic spline to capture any non-linear, seasonal patterns in outcomes, such as the impact of activities related to the annual World NEC Awareness Day[29] and International Kangaroo Care Awareness Day[30] (both held annually in May), or any variation in NEC incidence related to seasonality of circulating pathogens.[31]

► To account for any potential impact of the COVID-19 pandemic, we will include a binary variable coded as 1 during periods of national 'lockdown' and 0 otherwise, in line with our previous work examining the

impact of pandemic-related visitor restrictions on own mother's milk feeding.[32]

► ITS studies are not usually affected by confounders relating to population characteristics as these do not typically change significantly over time.[33] If the characteristics of infants born during each of the three study time periods differ, we will include covariates to capture any differences in potential confounders by month of birth, such as: median GA; percentage of male admissions; median birth weight; percentage of infants with a high-risk NMR-2000 score (a validated score of in-hospital mortality risk)[34]; percentage of infants with a major congenital anomaly; and median length of stay.

► Plots of the autocorrelation function and partial autocorrelation function will be examined to identify plausible values of the autoregressive (AR) and/or moving average (MA) terms to capture autocorrelation between data points. We will use the AIC to compare models with different plausible values of AR and MA terms, with a smaller AIC indicating better model fit.

### Multiple testing

IRRs for the primary and secondary outcomes will be reported with 95% CIs and Bonferroni-corrected p values.

### Secondary analyses

In addition to the ITS analysis, we will use multivariable logistic regression to conduct a conventional before-after comparison of the odds of each primary and secondary outcome, adjusted for a range of potential confounders measured at the level of the infant, mother and unit. We will include data from both EMNODN and control units and assess whether there is any effect modification by study group. We will use a robust variance estimator to account for the clustering of infants within units and by mother.

### Missing data

We will describe the amount of missing data for covariates. For the multivariable logistic regression, we will assume that any missing data occur randomly between the study phases and intervention and control groups and will impute missing values 10-fold using chained equations. Results will be pooled according to Rubin's rule.

### Data analysis strategy: workstream 2

We will use the RE-AIM framework,[14 15] supplemented by the CFIR[16] and the FIF,[17] to examine implementation of the care bundle, mechanisms of impact and the impact of contextual factors, to map the 'how' and 'why' underpinning the results of the quantitative evaluation undertaken in workstream 1. These frameworks bring together metrics for evaluating implementation success, and by assessing reach and effectiveness the frameworks can be used together to identify ways to produce sustained changes (RE-AIM and FIF), which will highlight modifiable factors that explain and enhance implementation outcomes (CFIR). We will triangulate data from the

NNRD, case report forms, and parent and HCP questionnaires and interviews to address the five evaluative dimensions of RE-AIM and map the CFIR constructs and FIF checklist within these domains (as appropriate; eg, particularly around Adoption, Implementation and Maintenance), to understand factors impacting implementation:

1. Reach: We will present information on numbers and percentages of infants and parents exposed to different elements of the care bundle and any exceptions or exclusions, considering variation by socioeconomic and demographic characteristics. We will measure numbers of HCPs trained to deliver the care bundle and perceptions of the acceptability and usefulness of training.

2. Efficacy: In addition to the overall efficacy assessment in workstream 1, we will measure the efficacy of key components of the care bundle and investigate their impact on individual-level outcomes. For example, we will compare rates of own mother's milk feeding between mothers who received antenatal breast feeding counselling and mothers who did not.

3. Adoption: With HCPs and unit leads, we will explore the contextual factors that vary between units that may impact adoption and fidelity of recommended practices, using the FIF checklist and CFIR to guide thematic coding. Moderators of adoption and fidelity may include staff training and existing expertise, level of available resource locally.

4. Implementation: We will explore factors that may affect implementation of the care bundle using CFIR as a coding framework. This will include factors related to the care bundle (eg, its complexity and evidence base), factors related to setting (eg, staff capacity to deliver the care bundle, availability of resources, time and expertise and leadership buy-in), attributes of roles within the teams and their motivations to implement the care bundle, and if and how the care bundle is adapted in each setting. This will help us better understand barriers and facilitators to implementation of the bundle.

5. Maintenance: We will assess the extent to which the care bundle is embedded in training, policy and guidance.

Interview data will be analysed using the framework approach,[35 36] a hierarchical, matrix-based method to map thematic differences/similarities within and between groups. First, a priori codes relating to the RE-AIM domains (Adoption, Implementation and Maintenance) will be applied to the data. Transcripts will then be coded again to apply the CFIR domains, constructs or subconstructs and FIF checklist (identified to influence RE-AIM domains), inductively. As an initial step to check validity of coding, two researchers will compare coding of several transcripts to reach consensus on interpretations. The remaining data will then be coded according to the relevant RE-AIM, CFIR and FIF code domains and construct codes, and the generated themes, with matrices of results and quotes presented and discussed among the

full research team, to agree on data interpretation and the final themes.

## Sample size calculation

Workstream 1: The ITS analysis will have approximately 90% power to detect an absolute increase of 20% in the prevalence of receiving own mother's milk at discharge and 80% power to detect a 15% absolute increase (baseline prevalence approximately 55%, approximately 23 infants per month in the EMNODN from 2017 to 2019; national average c.60%). This target is comparable to the prevalence of own mother's milk feeding in the best performing neonatal network—in 2021, the South London Neonatal Operational Delivery Network achieved a prevalence of 76% (range by unit 71–80%) of infants receiving own mother's milk at discharge.[37] In 21 neonatal units in the UK in 2021, over 75% of infants received own mother's milk at discharge, with the highest performing unit achieving 86% prevalence.[37]

NEC is a rare occurrence (incidence approximately 7%, approximately 3 cases per month in the EMNODN from 2017 to 2019; national average c.5%), and there is also considerable monthly variability in incidence. Therefore, statistical power to detect small changes in the incidence of NEC will be more limited; we will be powered to detect approximately a two-thirds reduction in the incidence of NEC. In line with previous studies of less common neonatal outcomes,[27] the point estimate will, however, give an indication of the effect of the care bundle, even if the CI around this point estimate crosses the null value.

Workstream 2: We aim to conduct 33 interviews with parents (either individuals or couples, 3 per EMNODN unit) and 33 interviews with HCPs (3 per unit) using a combination of convenience (self-selecting) and purposeful sampling approaches, the latter particularly to approach parents from underserved groups who usually hesitate to participate in research. These numbers are based on principles of maximum variation and theoretical data saturation, which is related to the sampling approach rather than the analysis.[38] Preliminary analysis will inform whether we need to use a second strategy based on the notion of inductive thematic saturation[39] and, in which case, we may need to conduct additional interviews.

## PATIENT AND PUBLIC INVOLVEMENT

The study protocol has been co-produced with the EMNODN Parent Advisory Group and the parent-led charity NEC UK[40] to ensure representation from those with lived experiences of NEC and its consequences, both in the neonatal unit and after discharge. The chair of NEC UK is a parent co-applicant and will lead patient and public involvement (PPI) activities. A second PPI representative will also be invited to join the Project Management Group.

The primary outcome of receiving own mother's milk feeds at discharge was selected by the parents as being the most meaningful outcome for them, and, further to their advice, the emphasis on receiving information and support for breastmilk expression in the process evaluation and the plans to explore this in detail in the semistructured parent interviews were included in the study plan.

With the support of The Centre for Ethnic Health Research,[41] a semistructured focus group discussion has been conducted with women representing different ethnic minority groups, all of whom had experience of having a preterm baby. This discussion focused on co-production of study documentation, particularly the participant information sheets and interview guides. A semistructured focus group discussion has also been conducted with HCPs working with preterm infants in the East Midlands. This discussion focused on the production of the interview guide for HCPs. Interested parties from both focus groups will be invited to validate qualitative analyses (respondent validation), comment on results and recommendations and review and revise the public-facing dissemination materials. A final parent-led dissemination meeting will be held within the EMNODN.

## ETHICS AND DISSEMINATION

The study is sponsored by the University Hospitals of Derby and Burton NHS Foundation Trust and has been registered on the ClinicalTrials.gov database (Ref: NCT05934123). The study has been approved by the South East Scotland Research Ethics Committee 01 and the Health Research Authority and Health and Care Research Wales (IRAS 323099). We will use the Reporting of Studies using Observational Routinely Collected Data (RECORD) checklist,[42] an extension of the Strengthening the Reporting of Observational Studies in Epidemiology guidelines,[43] to ensure transparency of reporting of our methods and results. On completion of the study, findings will be disseminated via: a detailed report for National Institute for Health and Care Research as well as the EMNODN and participating neonatal units; public-facing materials (infographics, animated video) in English, Polish, Gujarati and Punjabi (the most frequently spoken languages in the East Midlands), created and shared via parent groups and directly with study participants; open access, peer-reviewed journals and scientific conferences; and dissemination to relevant specialised commissioners including the NHS England Neonatal Critical Care Clinical Reference Group.

## DISCUSSION

National recommendations cite a need for QI efforts to increase own mother's milk feeding to reduce NEC.[44] In 2020, NNAP recommended that if rates of NEC are relatively high, neonatal networks and units should 'seek to identify and implement potentially better practices' and to 'focus on the early initiation and sustainment of breastmilk feeding in conjunction with parents by reviewing

data and processes' and 'undertake selected QI activities suited to the local context'.[44] This multicomponent evaluation of the care bundle developed for use in the East Midlands will provide evidence to help understand whether the bundle should be rolled out nationally and what changes might be needed to ensure optimum impact and successful implementation within local contexts. Thus, the findings of this study have the potential to improve patient care in the EMNODN and wider, with direct benefits to preterm infants and their families. The parallel effectiveness and process evaluations will establish a framework and toolkit for evaluation of other similar QI initiatives in future.

**Acknowledgements** We acknowledge the East Midlands Neonatal Operational Delivery Network (EMNODN) team for their support in setting up the study. We also acknowledge the authors of the EMNODN Care Bundle (Dr Dushyant Batra, Dr Deepa Panjwani, Dr Lara Shipley, Dr Chiara Taylor, Dr Rizma Aishath Moosa and Ms Ruth Prigg) for their contribution to the creation of the care bundle. Electronic patient data recorded at participating neonatal units that collectively form the UK Neonatal Collaborative are transmitted to the Neonatal Data Analysis Unit to form the National Neonatal Research Database (NNRD). We are grateful to all the 19 families who agreed to the inclusion of their baby's data in the NNRD, the health professionals who recorded data and the Neonatal Data Analysis Unit team.

**Contributors** SO conceptualised and designed the study, contributed to analyses plan and will lead the delivery and reporting. LS planned the statistical analysis and MB and EO planned the mixed methods process evaluation. JA is the study manager and will perform the qualitative data collection and analysis. HB contributed to study design and will provide input into the interpretation and writing of the results. MS is the PPI lead and will lead the development of public-facing dissemination materials. All authors contributed to the overall study design, development of the protocol and the writing and review of this paper.

**Funding** This project is funded by the National Institute for Health and Care Research (NIHR) under its Research for Patient Benefit (RfPB) programme (Grant Reference No NIHR203590).

**Disclaimer** The views expressed are those of the author(s) and not necessarily those of the NIHR or the Department of Health and Social Care.

**Competing interests** SO is an honorary neonatal consultant at the University Hospitals of Derby and Burton and is the lead clinician in the team that developed the EMNODN NEC Care Bundle. HB is an honorary neonatal consultant at the Nottingham University Hospitals and was on the advisory panel of the team that developed the EMNODN NEC Care Bundle. MS is the chair of NEC UK which supports families with experience of NEC and supports breastfeeding mothers.

**Patient and public involvement** Patients and/or the public were involved in the design, or conduct, or reporting, or dissemination plans of this research. Refer to the Methods section for further details.

**Patient consent for publication** Not applicable.

**Provenance and peer review** Not commissioned; externally peer reviewed.

**ORCID iDs**
Lisa Szatkowski http://orcid.org/0000-0003-3295-5891
Manpreet Bains http://orcid.org/0000-0002-1990-5948
Shalini Ojha http://orcid.org/0000-0001-5668-4227

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
