## [Reviewer comments · BMJ Open]

ARTICLE DETAILS

TITLE (PROVISIONAL)	Effects of implementation of a care bundle on rates of necrotising enterocolitis and own mother's milk feeding in the East Midlands: protocol for a mixed methods impact and process evaluation study
AUTHORS	Abramson, Janine; Szatkowski, Lisa; Bains, Manpreet; Orton, Elizabeth; Budge, Helen; Spruce, Marie; Ojha, Shalini

VERSION 1 – REVIEW

REVIEWER	Gephart, Sheila The University of Arizona, College of Nursing
REVIEW RETURNED	11-Oct-2023

GENERAL COMMENTS	This research protocol outlines an important quality improvement target with clarity. Stakeholders were engaged in the selection of the research question, primary outcome, and development of interview guides and dissemination planning. These few comments are addressible weaknesses: 1. Consider a subgroup analysis that breaks out those who received probiotics from those who did not.2. How with the framework for implementation fidelity be operationalized? Will you use a checklist or multiple auditors? More detail would be helpful.3. Description of using the CFIR to evaluate the implementation is broad and includes few details.4. Overall, this reads well with a few exceptions. In the abstract, check the grammar and revise this sentence: "In workstream 1, IS a retrospective cohort..." In the article summary, change "unpick the 'how'.." to "unpack..."5. Given the seasonality of NEC, I am concerned with only including 9 months vs. 12 of post-implementation data.6. In conducting the process evaluation, please consider capturing nursing data related to hours per patient day, average years of experience, and the availability of lactation consultants or peer counselors in the units. Nicely done! I look forward to reading the results once you complete your project!
---

REVIEWER	Victoria, C Federal University of Pelotas
REVIEW RETURNED	16-Oct-2023

GENERAL COMMENTS	This is a simple, yet well-designed and relevant study that combines implementation research with impact analyses. The latter may end up being limited by sample size issues, as the authors recognize – particularly regarding NEC incidence. The
--

	proposed design, including interrupted time series analyses, is appropriate. I felt the need for a descriptive section at the very beginning of the proposal. Some of the information I would like to see is provided later in the proposal, but it would be helpful to have such a description early on. The descriptive section could include the following aspects. How large is the EMODN in area and population? How many births a year are there (there is a mention to 23 births/month later in the proposal)? Are the 11 NICUs all the existing ones in the health authority? If not, how were these selected? What was the baseline prevalence (not incidence, as stated) of exclusive breastfeeding at discharge, and how does it compare to the rest of the country (we are only told that it was lower in EMODN)? In the sample size section we are informed that the incidence of NEC in the EMODN was 7%, but how does this compare to the rest of the country? What would be a realistic estimate of the impact of the program (e.g. by comparing pre-intervention incidence rates with those in other ODNs in the country with effective programs)? Other than this request for more information up front, the authors did a good job in presenting the proposal for a mostly solid and relevant study.
--	--

VERSION 1 – AUTHOR RESPONSE

Reviewer: 1	
1. Consider a subgroup analysis that breaks out those who received probiotics from those who did not.	Thank you for this suggestion. Only one unit in the East Midlands Neonatal Operational Delivery Network uses probiotics, and therefore we would have limited power to conduct a subgroup analysis comparing those who do and do not receive probiotics.
2. How will the framework for implementation fidelity be operationalized? Will you use a checklist or multiple auditors? More detail would be helpful.	We have added further detail on the use of FIF, alongside further information on use of CIFR and RE-AIM, in the description of our data analysis strategy. We will use the FIF checklist to guide our coding of interview data pertaining to adoption and fidelity of the care bundle. Two researchers will initially compare coding of several interview transcripts to reach consensus on interpretation, followed by discussion among the full research

	team to agree on data interpretation and the final themes. Please note that the three Frameworks will be used together to guide the analysis. We have amended parts of the methods to make this clearer.
3. Description of using the CFIR to evaluate the implementation is broad and includes few details.	We have amended parts of the Methods (data analysis strategy) section to incorporate more details on how CFIR will be used to evaluate implementation. The analysis section has also been amended to outline how RE-AIM, CFIR and FIF will be used to analyse the data and bring the three frameworks together.
4. Overall, this reads well with a few exceptions. In the abstract, check the grammar and revise this sentence: "In workstream 1, IS a retrospective cohort..." In the article summary, change "unpick the 'how'.." to "unpack..."	We have amended these sentences as advised and have made a small number of minor changes to grammar throughout.
5. Given the seasonality of NEC, I am concerned with only including 9 months vs. 12 of post-implementation data.	Thank you for your comment. We already plan to include month of the year as a cyclic cubic spline to capture any non-linear, seasonal patterns in outcomes, such as the impact of activities related to the annual World NEC Awareness Day and International Kangaroo Care Awareness Day (see section on Data analysis strategy for workstream 1). This term will also capture any seasonality related to, for example, ambient temperature. We have added a reference to this section to acknowledge this potential seasonal variation. Our use of a quasi-Poisson generalised additive mixed model will model the number of cases of NEC on a monthly basis and so the fact that we only have 9 months of post-implementation data is not problematic. Only if we were conducting a simple before and after comparison of the number of cases would it be important to have a full year of post-implementation data.
6. In conducting the process evaluation, please consider capturing nursing data related to hours per patient day, average years of experience, and the availability of lactation consultants or peer counselors in the units.	Thank you for these suggestions. In our interviews with HCPs and unit leads we already plan to explore the contextual factors which vary between units that may impact adoption and fidelity of recommended practices e.g., cost, level of resource, knowledge, training, staff expertise (see section on Data analysis strategy for workstream 2). We have recently begun to conduct interviews and the points you raise

	have all been brought up by our first interviewees. We do indeed also plan to access staffing data for each EMNODN unit.
Reviewer: 2	
I felt the need for a descriptive section at the very beginning of the proposal. Some of the information I would like to see is provided later in the proposal, but it would be helpful to have such a description early on. The descriptive section could include the following aspects.	Thank you for this suggestion. We have woven your suggestions into the existing text in the introduction, and have given more details in our sample size calculation to place data from the EMNODN in context nationally.
How large is the EMODN in area and population?	In our introduction we now give figures for the size of the area and population covered by the EMNODN, plus the total number of live births in the region per year.
How many births a year are there (there is a mention to 23 births/month later in the proposal)?	As well as now giving data on the total number of live births in the region per year, we have also included an indication of the workload of EMNODN units based on the most recent available data from the National Neonatal Audit Programme: "In 2022, EMNODN units provided 50,443 days of care in total to infants born preterm, or unwell at term, who needed specialist neonatal care; 476 infants were cared for who were born before 32 weeks' gestational age and survived at least 48 hours."
Are the 11 NICUs all the existing ones in the health authority? If not, how were these selected?	The EMNODN comprises of 11 neonatal units which we have made clear in our description. There was no sampling of units for the purposes of this study.
What was the baseline prevalence (not incidence, as stated) of exclusive breastfeeding at discharge, and how does it compare to the rest of the country (we are only told that is was lower in EMODN)?	The baseline prevalence was approximately 55% as we already describe in the sample size section. The national average was approximately 60% for the comparable time period. We have added this detail to the manuscript.
In the sample size section we are informed that the incidence of NEC in the EMODN was 7%, but how does this compare to the rest of the country?	The national average was approximately 5% for the comparable time period. We have added this detail to the sample size section.
What would be a realistic estimate the impact of the program (e.g. by comparing pre-intervention	In the sample size section we write that the interrupted time series analysis will have approximately 90% power to detect an absolute

incidence rates with those in other ODNs in the country with effective programs)?

increase of 20% in the prevalence of receiving own mother's milk at discharge and 80% power to detect a 15% absolute increase. These magnitudes of effect are comparable to the impact of the care bundle being to improve the level of own mother's milk feeding to the level seen in the best performing neonatal network nationally (where 76% of infants receive own mother's milk at discharge).